# Selenium substitution for dielectric constant improvement and hole-transfer acceleration in non-fullerene organic solar cells

Xinjun He[1,9], Feng Qi[2,3,4,9], Xinhui Zou[5,9], Yanxun Li[2,3,6], Heng Liu[7], Xinhui Lu[7], Kam Sing Wong [5] ✉, Alex K.-Y. Jen [2,3,6,8] ✉ & Wallace C. H. Choy [1] ✉

Dielectric constant of non-fullerene acceptors plays a critical role in organic solar cells in terms of exciton dissociation and charge recombination. Current acceptors feature a dielectric constant of 3-4, correlating to relatively high recombination loss. We demonstrate that selenium substitution on acceptor central core can effectively modify molecule dielectric constant. The corresponding blend film presents faster hole-transfer of ~5 ps compared to the sulfur-based derivative (~10 ps). However, the blends with Se-acceptor also show faster charge recombination after 100 ps upon optical pumping, which is explained by the relatively disordered stacking of the Se-acceptor. Encouragingly, dispersing the Se-acceptor in an optimized organic solar cell system can interrupt the disordered aggregation while still retain high dielectric constant. With the improved dielectric constant and optimized fibril morphology, the ternary device exhibits an obvious reduction of non-radiative recombination to 0.221 eV and high efficiency of 19.0%. This work unveils heteroatom-substitution induced dielectric constant improvement, and the associated exciton dynamics and morphology manipulation, which finally contributes to better material/device design and improved device performance.

Organic solar cells (OSCs) are promising candidate for clean energy application due to the exceptional advantages such as esthetic feature, tunability for chemical structure, and solution process[1,2]. Due to the vast development of non-fullerene acceptors (NFAs), OSCs are able to absorb near-infrared light without sacrificing the output voltage, leading to a significantly improved power conversion efficiency (PCE) to over 19%[3–8]. Further improvement of PCE is limited by the relatively low dielectric constant ($\varepsilon_r$) of the organic semiconductors.

$\varepsilon_r$ is the relative permittivity of a substance to vacuum, which describes the capability of a material to contain an electric flux. In the very early stage of photovoltaic process in a solar cell, a Coulombically bounded electron–hole pair (exciton) will be generated upon solar illumination, which shall be separated into free carriers. The exciton binding energy ($E_b$) is highly sensitive to $\varepsilon_r$. There are reports concluded the relationship between $E_b$ and $\varepsilon_r$ in different OSC system as $E_b \propto 1/\varepsilon_r$[9]. When $\varepsilon_r$ increases from 3 to 4, $E_b$ reduces by about 80 meV in certain OSC system[10], which contributes to higher kinetic energy of the

[1]Department of Electrical and Electronic Engineering, The University of Hong Kong, Pokfulam Road, Hong Kong SAR, China. [2]Department of Chemistry, City University of Hong Kong, Kowloon, Hong Kong. [3]Hong Kong Institute for Clean Energy (HKICE), City University of Hong Kong, Kowloon, Hong Kong. [4]College of Materials Science and Engineering, Qingdao University, Qingdao, P. R. China. [5]Department of Physics and William Mong Institute of Nano Science and Technology, The Hong Kong University of Science and Technology, Clear Water Bay, Hong Kong SAR, China. [6]Department of Materials Science and Engineering, City University of Hong Kong, Kowloon, Hong Kong. [7]Department of Physics,  Chinese University of Hong Kong, New Territories, Hong Kong SAR, China. [8]Department of Materials Science and Engineering, University of Washington, Seattle, WA, USA. [9]These authors contributed equally: Xinjun He, Feng Qi, Xinhui Zou. ✉e-mail: phkswong@ust.hk; alexjen@cityu.edu.hk; chchoy@eee.hku.hk

exciton/carrier, and thus accelerates exciton dissociation and charge transfer. Typical inorganic solar cell semiconductors exhibit higher $\varepsilon_r$ such as Si (-12)[11] and Gallium Arsenide (-12)[12,13], the emerging organic/inorganic hybrid perovskite solar cells present $\varepsilon_r$ of around 30[14], leading to spontaneous charge separation upon exciton generation. In contrast, the OSC semiconductors normally exhibit $\varepsilon_r$ of 3–4 (see refs. 15–19), which is one of the main reasons to the relatively high non-radiative recombination loss in OSCs. Notably, some of them can reach an $\varepsilon_r$ over 5 by attaching polarizable side chain such as oligo(ethylene glycol) and cyano groups on the organic backbone[15,20,21], whereas this may significantly alter the molecular geometry and solubility.

Selenium substitution of sulfur in the thiophene ring of NFAs has been widely adopted to extend the light absorption range to near-infrared region[22–26]. The lower electronegativity (2.4) and larger atom radius enable Se a more polarizable surface, which contributes to stronger intermolecular interactions of the NFAs[27–30]. It draws our attention that replacing the sulfur with more electron-donating Se at the outer thiophene ring of the dithienothiophen[3,2-b]-pyrrolo-benzothiadiazole NFA core would increase the dipole moment of the molecule, periodic stacking of the molecules will enhance the overall dipole and finally improve the bulk $\varepsilon_r$. Notably, compared to conventional strategies to modify $\varepsilon_r$ by attaching polar segment on the backbone and adding extra polar substance in the organic film[15], Se substitution on the central core does not significantly change the molecular structure and geometry.

In this contribution, we initiate the investigation of $\varepsilon_r$ adjustment brought by Se substitution in NFA, and the related electrical and morphological evolution. Double Se substitution at the outermost thiophene ring of NFA central core induced an obvious increase of dipole moment. The corresponding bulk Se-NFA film, T9SBO-F[25], presents obviously enhanced $\varepsilon_r$ of 5.04. When blended with polymer donor, the hole-transfer time reduces from ~10 ps to about ~5 ps, which is mainly due to the enhanced $\varepsilon_r$ and reduced $E_b$. Surprisingly, the Se-based blend exhibits faster charge recombination after hundreds of picosecond upon excitation, which is counterintuitive because it is more difficult for the separated charges to recombine in higher $\varepsilon_r$ materials. This is ascribed to the relatively disordered packing of T9SBO-F domain. The disordered domain increases the recombination chances for the dissociated charges. Therefore, the device based on

PM6:T9SBO-F presents a PCE of 18.4%. Encouragingly, dispersing T9SBO-F into another OSC framework, PM6:L8-BO (full name shown in "Methods"), helps to regulate its packing behavior while still preserves the enhanced $\varepsilon_r$, leading to shortened hole-transfer time of ~7 ps together with suppressed bimolecular recombination. The resulted PM6:L8-BO:T9SBO-F device delivered a PCE of 19.0% with a reduced non-radiative recombination loss of 0.221 eV. Overall, design high $\varepsilon_r$ NFAs without changing its molecular geometry, and concurrently manipulate their morphology paves an avenue for high-performance OSCs.

## Results

### Dielectric constant and photovoltaic performances

First, we have simulated the electrostatic potential distribution (ESP) of the NFA cores with Se substitution on different positions (Supplementary Fig. 1). The geometry is optimized by Gaussian at B3LYP (Becke, 3-parameter, Lee–Yang–Parr)/6-31 G (d, p) level[31]. Without Se substitution, the NFA core presents a dipole moment of 5.68 Debye. Se substituted on the inner and middle thiophene ring of dithie-nothiophen[3,2-b]-pyrrolobenzothiadiazole NFA core will lead to a decrease of dipole moment while Se substitution on the outer position favors the increase of dipole moment. Finally, double Se substitution in the outer position leads to the most prominent increase to 6.04 Debye. This is attributed to that Se features a stronger electron-donating property compared to sulfur, which also enables stronger intra-molecular interactions. We then calculate the ESP of L8-BO (Fig. 1a) and the double Se-substituted NFA, (2,10-bis(2-methylene-(3-(1,1-dicyanomethylene)-5,6-difluoroindanone))-12,13-bis(2-butyloctyl)-3,9-dinonyl-diselenopheno[2″,3″:4′,5′]thieno[2′,3′:4,5]pyrrolo[3,2-e:2′,3′-g][2,1,3]benzothiadiazole), namely T9SBO-F (Fig. 1b, c)[25]. Both L8-BO and T9SBO-F almost show positive ESP distribution on the iso-surface around the backbone. L8-BO molecule presents a dipole moment of 0.57 D, whereas it is significantly increased to 3.26 D in T9SBO-F. The highest occupied molecular orbital (HOMO) and lowest unoccupied molecular orbital (LUMO) energy levels are characterized to be −5.64 and −3.87 eV, respectively, by cyclic voltammetry method (Supplementary Fig. 2a). The T9SBO-F neat film shows broad light absorption from 600 to 950 nm with the peak centered at 846 nm (Supplementary Fig. 2b).

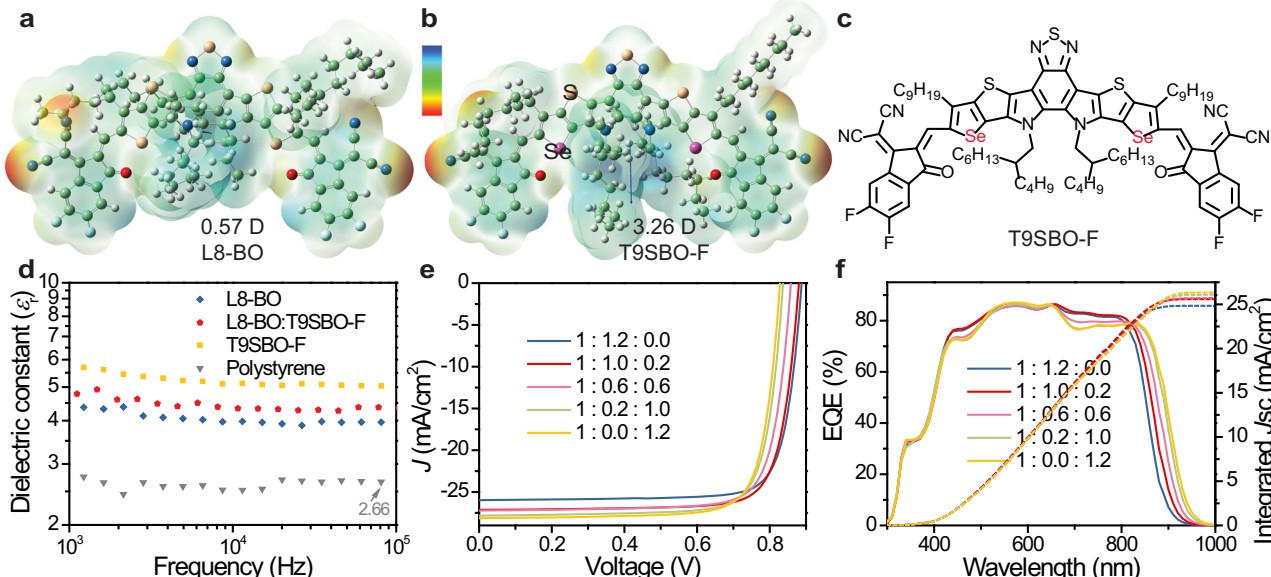

**Fig. 1 | Chemical structure, electrical and photovoltaic properties.** The electrostatic surface potential contour of NFA **a** L8-BO and **b** T9SBO-F. The scale bar is from −1.28 (red) to 1.28 eV (blue), D means Debye. **c** Chemical structure of T9SBO-F. **d** $\varepsilon_r$ of the organic films. **e** $J$–$V$ curve and **f** external quantum efficiency (EQE) spectra of the devices. The ratio shown in the Figures is PM6:L8-BO:T9SBO-F mass ratio. Source data are provided as a Source Data file.

Such broadband absorption can effectively capture near-infrared photons.

We probe $\varepsilon_r$ of the acceptor materials by electrochemical impedance spectroscopy, detailed description is provided in Supplementary Note I. Meanwhile, polystyrene (PS) as a widely employed dielectric layer in transistors has also been investigated for comparison[32,33]. $\varepsilon_r$ is sensitive to the frequency of AC modulation. There are reports concluded the dielectric response modes at different AC frequencies, where the dielectric response at $<10^6$ Hz originates from electronic and/or ionic space charges[34]. In addition, at $10^3$–$10^5$ Hz, the organic layer shall be fully depleted, and the measured $\varepsilon_r$ reflects the properties of the whole active layer[14]. As shown in Fig. 1d, PS presents an $\varepsilon_r$ of ~2.6 at $10^3$–$10^5$ Hz, which is consistent with the reported value[35,36]. Compared to L8-BO (3.96), T9SBO-F exhibits an obviously improved $\varepsilon_r$ of 5.04. Adding T9SBO-F into L8-BO with a ratio of 0.2:1 also improve the $\varepsilon_r$ to 4.37.

We then blend the acceptors with PM6 to fabricate conventional structured devices. The corresponding absorption and PL spectra of the organic films are shown in Supplementary Fig. 3. $\varepsilon_r$ of the blends with different acceptor ratios are measured in Supplementary Fig. 4. With the increase of T9SBO-F ratio in the acceptor, $\varepsilon_r$ increases from 3.36 (0%), to 3.51 (16.7%), 3.56 (50%), 3.93 (83.3%) and 4.15 (100%). The $J$–$V$ curves, EQE spectra and the device parameters with different acceptor ratios are listed in Fig. 1e, f, and Table 1, respectively. With the increase of T9SBO-F ratio in the acceptor, the device $Voc$ gradually decreases from 0.888 V (0%), to 0.881 V (16.7%), 0.858 V (50%), 0.837 V (83.3%), and 0.829 V (100%). While increasing the T9SBO-F ratio helps to extend the absorption to near-infrared region, the EQE value keeps decreasing in 650–800 nm, which is ascribed to the increased charge recombination and will be discussed in the next section. Overall, the $Jsc$ gradually increases from 26.0 mA/cm$^2$ (0%), to 27.1 mA/cm$^2$ (16.7%), 27.2 mA/cm$^2$ (50%), 27.8 mA/cm$^2$ (83.3%) and 28.1 mA/cm$^2$ (100%). When blending PM6:L8-BO:T9SBO-F at a ratio of 1:1:0.2, the device presents the highest PCE of 19.0% with a decent FF of 79.6%.

It is expected that improving $\varepsilon_r$ may contribute to the electrical properties of the blend. Therefore, we measured the carrier motilities of devices with different ratios and thicknesses. In the thin-layer devices (100 nm, Supplementary Fig. 5 and Supplementary Table 1), all the blend shows similar $\mu_h$ of around $10.5 \times 10^{-4}$ cm$^2$ V$^{-1}$ s$^{-1}$ and $\mu_e$ of $10 \times 10^{-4}$ cm$^2$ V$^{-1}$ s$^{-1}$, correlating to a balanced $\mu_h/\mu_e$. However, in the thick OSCs (~300 nm), PM6:T9SBO-F shows a significantly boosted $\mu_e$ of $40.7 \times 10^{-4}$ cm$^2$ V$^{-1}$ s$^{-1}$ compared to PM6:L8-BO ($12.4 \times 10^{-4}$ cm$^2$ V$^{-1}$ s$^{-1}$) and PM6:L8-BO:T9SBO-F (1:1:0.2, $13.6 \times 10^{-4}$ cm$^2$ V$^{-1}$ s$^{-1}$), while they presents similar $\mu_h$ of around $8 \times 10^{-4}$ cm$^2$ V$^{-1}$ s$^{-1}$, which renders an obviously unbalanced $\mu_h/\mu_e$ (0.17 in PM6:T9SBO-F) and decreased FF (Supplementary Fig. 6 and Supplementary Table 2). Note that the mobility will also be influenced by the morphology, we will present the discussion in later morphology section.

**Selenium substitution impact on exciton/charge dynamics**

To simplify the description in the following discussion, we select the best-performed ternary device (PM6:L8-BO:T9SBO-F of 1:1:0.2) to compare with the binary devices. To understand the exciton/charge dynamics, we carry out ultrafast transient absorption (TA) spectroscopy on the organic films. The TA spectra and corresponding 2D color plots are displayed in Supplementary Fig. 7 and Fig. 2a–c, respectively. The detailed spectral features are described in Supplementary Note III. In brief, upon optical pumping the organic blends at 800 nm, a local-exciton (LE) signal (900–950 nm) first emerges, followed by an intermediate delocalized state (1300–1500 nm) and finally separated charge signal (CS, 750 nm) appears[37].

For the excited-state absorption (ESA) band at 1300–1500 nm, there are reports evidenced that this state helps to mediate the pathway for free-carrier formation, which is referred as delocalized singlet exciton state (DSE)[25,38]. Instead of the charge-transfer state that exists at the acceptor/donor interface, this DSE state spontaneously forms after LE state inside the properly stacked NFA acceptors (LE → DSE), which is because of strong hybridization between the excited species[38]. This is also seen in our work that a broad ESA signal emerges at 1300–1500 nm after the primary excitation of the blend and neat acceptor films (Supplementary Fig. 8). Based on the above result, we would expect that for a NFA with higher $\varepsilon_r$, it would exhibit accelerated hole-transfer properties because the enhanced localized electric field will assist the exciton dissociation. We then plot the ground-state bleach signal of the donor (GSB$_D$) to investigate the hole transfer in the blends. From the rising kinetics in Fig. 2e, we can identify two time-scales for hole transfer. We consider the faster dynamics ($\tau_1$) is because of the dissociation of the excitons near the donor/acceptor interface while the slower dynamics ($\tau_2$) is mainly due to the diffusion of the exciton/hole in the bulk acceptor domains. The fitting results of a biexponential model is summarized in Supplementary Table 3. PM6:T9SBO-F shows a $\tau_1$ of 0.49 ps and $\tau_2$ of 4.83 ps, which is almost half of the value in PM6:L8-BO ($\tau_1$ of 1.18 ps and $\tau_2$ of 9.15 ps), suggesting an accelerated hole transfer in the Se-substituted blends. Notably, the ternary film also presents faster dynamics with $\tau_1$ of 0.89 ps and $\tau_2$ of 6.21 ps than PM6:L8-BO. A similar phenomenon is observed during polaron formation (probed at 1000 nm, Fig. 2f), PM6:T9SBO-F and the ternary blend present faster-rising kinetics after 200 fs than that of PM6:L8-BO, suggesting faster polaron formation.

Despite the fast hole transfer and polaron formation in the Se-substituted blend, it suffers from faster recombination, which is evidenced by the fastest decay after 25 ps in the polaron kinetics (1000 nm, Fig. 2f) and >100 ps in the delocalized state (1300–1500 nm, Fig. 2d) in the PM6:T9SBO-F film. This is unexpected because the charge recombination barrier is higher in high $\varepsilon_r$ film. Notably, at the mentioned time scale, charge separation has already happened. Therefore, the faster recombination is ascribed to bimolecular recombination and is probably due to the poor film morphology. We will investigate the reasons in the next two sections. Importantly, the ternary blend (PM6:L8-BO:T9SBO-F) shows the slowest decay, indicating suppressed recombination.

We have also pumped the sample at 550 nm to selectively excite the donor polymer and to investigate electron transfer. Upon optical pumping, a primary bleach band centered at 640 nm is observed, correlating to GSB$_D$. Simultaneously, the broad ESA signal at 1050–1400 nm reflects the LE state in PM6. As shown in Fig. 2g and

**Table 1 | Summary of device photovoltaic parameters**

| PM6:L8-BO:T9SBO-F | $J_{SC}$ (mA/cm$^2$) | $J_{SC, EQE}$ (mA/cm$^2$) | $V_{OC}$ (V) | FF (%) | PCE (%) |
|---|---|---|---|---|---|
| 1:1.2:0 | 26.0 (25.9 ± 0.2) | 24.8 | 0.888 (0.888 ± 0.002) | 80.4 (79.9 ± 0.4) | 18.5 (18.3 ± 0.1) |
| 1:1.0:0.2 | 27.1 (26.8 ± 0.2) | 25.6 | 0.881 (0.877 ± 0.002) | 79.6 (79.8 ± 0.4) | 19.0 (18.8 ± 0.1) |
| 1:0.6:0.6 | 27.2 (27.0 ± 0.2) | 25.7 | 0.858 (0.858 ± 0.002) | 79.1 (78.9 ± 0.2) | 18.5 (18.3 ± 0.1) |
| 1:0.2:1.0 | 27.8 (27.7 ± 0.1) | 26.1 | 0.837 (0.836 ± 0.002) | 79.0 (78.5 ± 0.3) | 18.4 (18.2 ± 0.1) |
| 1:0:1.2 | 28.1 (27.9 ± 0.2) | 26.4 | 0.829 (0.827 ± 0.002) | 78.9 (78.9 ± 0.2) | 18.4 (18.2 ± 0.1) |

The average value is obtained from ten individual devices. Source data are provided as a Source Data file.

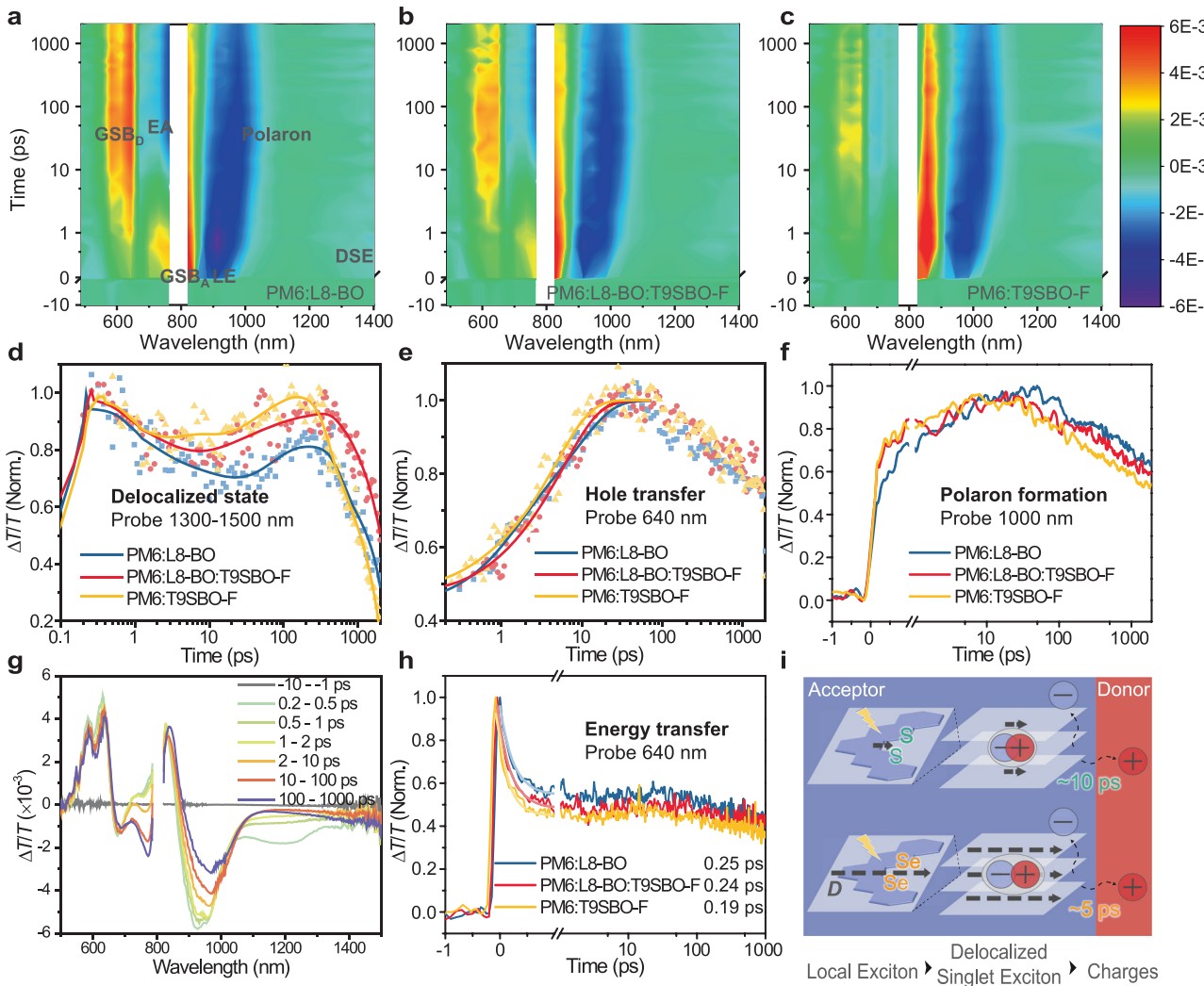

**Fig. 2 | Ultrafast femtosecond TA results of the organic films.** TA contour plots by exciting the acceptors at 800 nm of **a** PM6:L8-BO, **b** PM6:L8-BO:T9SBO-F, and **c** PM6:T9SBO-F. The TA kinetics of the samples excited at 800 nm and probed at **d** 1300–1500 nm, **e** 640 nm, and **f** 1000 nm. **g** TA spectra of PM6:L8-BO:T9SBO-F upon the excitation of donor at 550 nm. **h** The TA kinetics of samples excited at 550 nm and probed at 640 nm. **i** Schematic illustration of the Se-substituted NFA accelerating hole transfer, D means dipole moment. Source data are provided as a Source Data file.

Supplementary Fig. 9, the LE signal vanishes in 1 ps, implying very fast energy transfer from PM6 to the acceptors. After 5 ps, an electro-absorption (EA) signal at 750 nm emerges and persists to over 2 ns due to charge separation. We have plotted the $GSB_D$ signal (640 nm) in Fig. 2h. A shortened decay lifetime of 0.19 ps is characterized for the PM6:T9SBO-F film compared to 0.25 ps of PM6:L8-BO, suggesting T9SBO-F also enables faster energy transfer from PM6 to the acceptor.

Based on the above results, we illustrate the possible working mechanism of the Se-substituted NFA in Fig. 2i. Double Se substitution at outer thiophene of NFA core increases the dipole moment of the NFA and the bulky organic film exhibits higher $\varepsilon_r$. Notably, the exciton/carrier kinetics is dominated by the kinetic energy ($K_E$) of the quasi-particles: $K_E = h\upsilon - \phi - E_b$, where $h\upsilon$ is the excitation energy, $\phi$ is the energy gap of photoexcitation and transitions between different states (LE state → DSE/CT state → CS state). $E_b$ can be approximated by the equation: $E_b \approx e^2/(4\pi\varepsilon_0\varepsilon_r R)$[9,39], where $e$ is the elementary charge, $\varepsilon_O$ is the vacuum dielectric constant and $R$ is the average distance of the electron and hole. Therefore, a higher $\varepsilon_r$ can reduce the exciton binding energy and enable the exciton/carrier with more kinetic energy, and finally facilitate the exciton dissociation and hole transfer. T9SBO-F features 20 meV less $E_b$ than L8-BO neat film, calculated by subtracting the optical gap ($E_g^{opt}$, absorption onset) from the

fundamental bandgap ($E_g$)[40], correlating to accelerated hole-transfer properties. Importantly, dispersing the Se-substituted NFA into anther OSC system also helps with the hole transfer.

**Energy loss quantification and analysis**

Although PM6:T9SBO-F shows faster hole transfer and energy transfer than PM6:L8-BO, it suffers from severe recombination as described above. We then quantify the recombination loss of different devices based on electroluminescence (EL) and Fourier transform photo-current spectroscopy EQE ($EQE_{ftps}$) measurement, which is displayed in Fig. 3a–c. The bandgap energy ($E_g$) is defined as the mean peak energy at the absorption edge of the distribution $P(E_g)$, details is in Supplementary Note IV. Then, the total device energy loss can be identified by the difference between $E_g$ and $qV_{oc}$: $E_{loss} = E_g - qV_{OC}$. PM6:L8-BO and PM6:T9SBO-F present an $E_{loss}$ of 0.552 and 0.551 eV, respectively (Supplementary Table 4). The ternary device shows slightly lower loss of 0.549 eV.

The total $E_{loss}$ can be quantified via detailed balance theory, and is contributed from three types of energy loss ($E_{loss} = \Delta E_1 + \Delta E_2 + \Delta E_3$)[41]. The first term ($\Delta E_1$) is the energy difference between bandgap and $qV_{oc}$ in the Shockley-Queisser limit ($qV_{OC}^{SQ}$), $\Delta E_1 = Eg - qV_{OC}^{SQ}$, which is inevitable for different kinds of solar cells. It depends on the bandgap

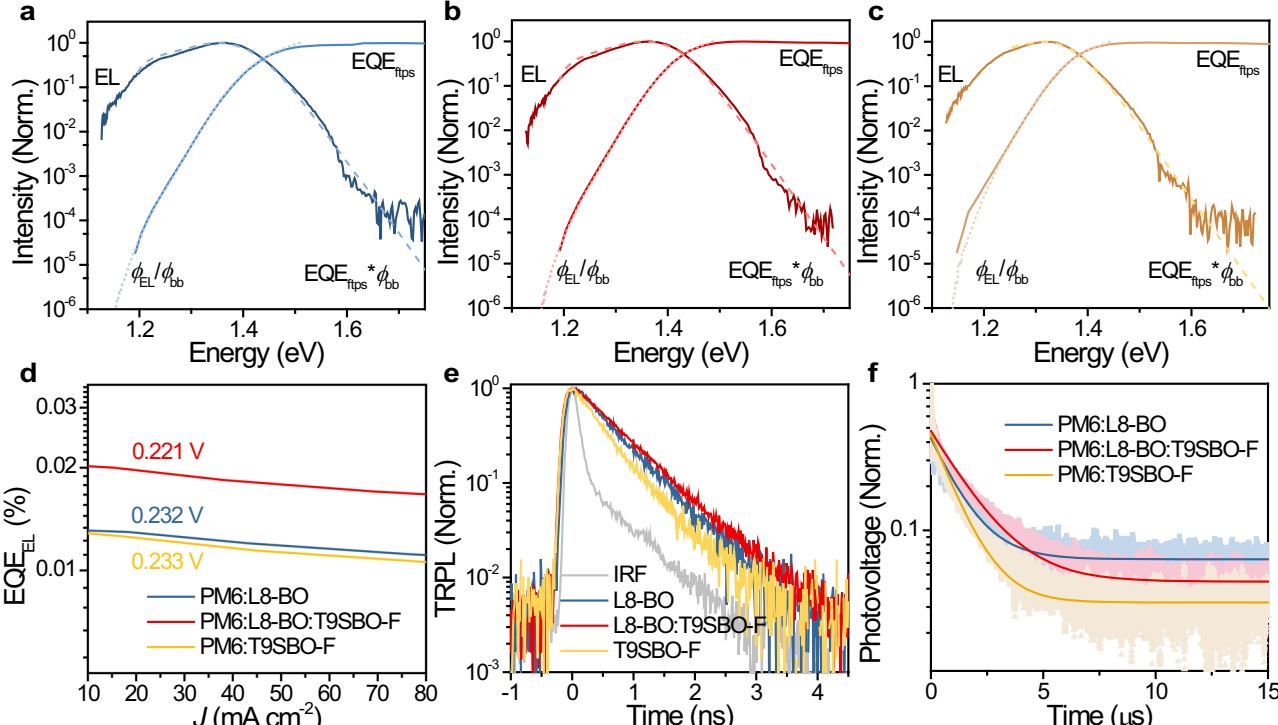

**Fig. 3 | Energy loss analysis of the OSC devices.** Semi-logarithmic plots of the normalized FTPS-EQE and electroluminescence (EL) spectra **a** PM6:L8-BO, **b** PM6:L8-BO:T9SBO-F, and **c** PM6:T9SBO-F. **d** EQE$_{EL}$, **e** time-resolved photoluminescence (TRPL), and **f** transient photovoltage characterizations of the devices. Source data are provided as a Source Data file.

and normally resides in 0.25 eV or above. Compared to 0.265 eV in PM6:L8-BO, PM6:T9SBO-F shows $\Delta E_1$ of 0.261 eV due to narrower bandgap. $\Delta E_2$ is the energy difference between the measured absorptance of a solar cell ($qV_{OC}^{rad}$) and that of SQ limit, which describes the radiative recombination loss below bandgap, $\Delta E_2 = qV_{OC}^{SQ} - qV_{OC}^{rad}$. 0.052, 0.059, and 0.052 eV is obtained for L8-BO, L8-BO:T9SBO-F and T9SBO-F-based devices, respectively. The third component, $\Delta E_3$, is ascribed to non-radiative recombination loss, which can be calculated from external radiative efficiency of EL ($\Delta E_3 = -k_B T \ln(\text{EQE}_{EL})$). This is also the dominating and challenging component that limits device performance[6]. As confirmed in Fig. 3d, the ternary device shows an obviously higher EQE$_{EL}$ of 0.019% than 0.013% for PM6:L8-BO and 0.012% for PM6:T9SBO-F. In other words, the binary Se-substituted solar cell shows the highest $\Delta E_3$ of 0.233 eV, followed by a $\Delta E_3$ of 0.232 eV of L8-BO device. Notably, adding T9SBO-F into PM6:L8-BO reduces the non-radiative recombination loss to 0.221 eV.

To supplement the recombination loss analysis, we also conducted other photoelectrical measurements. Figure 3e performs the time-resolved photoluminescence (TRPL) results of the neat acceptor samples. TRPL lifetime of 0.643, 0.717, and 0.509 ns is fitted for neat L8-BO, L8-BO:T9SBO-F and neat T9SBO-F film, respectively. The fastest decay suggests the most severe recombination in T9SBO-F sample. Blending L8-BO with T9SBO-F together helps to alleviate the recombination, thus the sample shows extended lifetime. Similarly in Fig. 3f of transient photovoltage (TPV) characterization, a TPV lifetime of 1.18, 1.60, and 1.06 μs is recorded for PM6:L8-BO, PM6:L8-BO:T9SBO-F and PM6:T9SBO-F devices, respectively. Since devices are connected with a 5.5 MΩ resistor in series, the populated carriers tend to recombine in the device. Longer lifetime indicates suppressed recombination in the ternary device. Transient photocurrent (TPC) lifetime is 0.323, 0.295, and 0.282 μs for PM6:L8-BO, PM6:L8-BO:T9SBO-F and PM6:T9SBO-F devices, respectively (Supplementary Fig. 11a). The shortest TPC lifetime in Se-based binary device is likely due to that a portion of the

carriers recombine instead of be extracted by the out circuit. We have also performed light intensity-dependant measurement (Supplementary Fig. 11). Consequently, T9SBO-F-based binary device shows severer recombination, whereas blending the two acceptors together helps to reduce the recombination.

**Molecular stacking behavior of neat acceptors and blend films**
In order to reveal the reasons for the recombination loss observed in last two sections, we performed morphological study of the samples. Grazing-incidence wide-angle X-ray scattering (GIWAXS) is performed on the blend and neat samples (Fig. 4a–f). The corresponding 1D line-cut profiles are plotted in Fig. 4g–j. The blend films show very similar scattering feature at around 0.31 Å$^{-1}$ in the in-plane (IP) and 1.76 Å$^{-1}$ in the out-of-plane (OOP) direction, correlating to lattice spacing ($d_{\text{spacing}}$) of 20.27 and 3.57 Å, which can be indexed to (100) lamellar stacking and (010) π-π stacking, respectively. Such scattering signal suggests a favored face-on orientation of the organic molecules and is beneficial for the vertical charge transportation in the devices. Compared to that, the neat acceptor films present a larger lamellar stacking of 0.36 Å$^{-1}$, corresponding to a reduced $d_{\text{spacing}}$ of 17.45 Å. Neat L8-BO film features a π-π stacking of 1.82 Å$^{-1}$, correlating to a $d_{\text{spacing}}$ of 3.46 Å, while T9SBO-F shows a π-π stacking of 1.73 Å$^{-1}$ and a $d_{\text{spacing}}$ of 3.62 Å. The increased OOP π-π stacking distance is likely due to the bigger atomic radius of Se (1.2 Å) compared to S (1.0 Å). Another consequence is that the T9SBO-F film exhibits several disordered lamellar scattering signals at 0.2–0.5 Å in IP direction (Fig. 4i). Previous reports have referred these peaks to different stacking information, e.g., the assignment of $Q_{xy}$ at 0.42 Å$^{-1}$ to end-group π–π stacking[4,42]. A similar scattering pattern is observed in the OOP direction at 0.4–0.8 Å (Fig. 4j). It is deduced that such disordered packing mode of T9SBO-F molecule is not beneficial for the final device performance.

To explain the "disordered stacking" behavior of T9SBO-F, we have successfully grown T9SBO-F single crystal, collected the

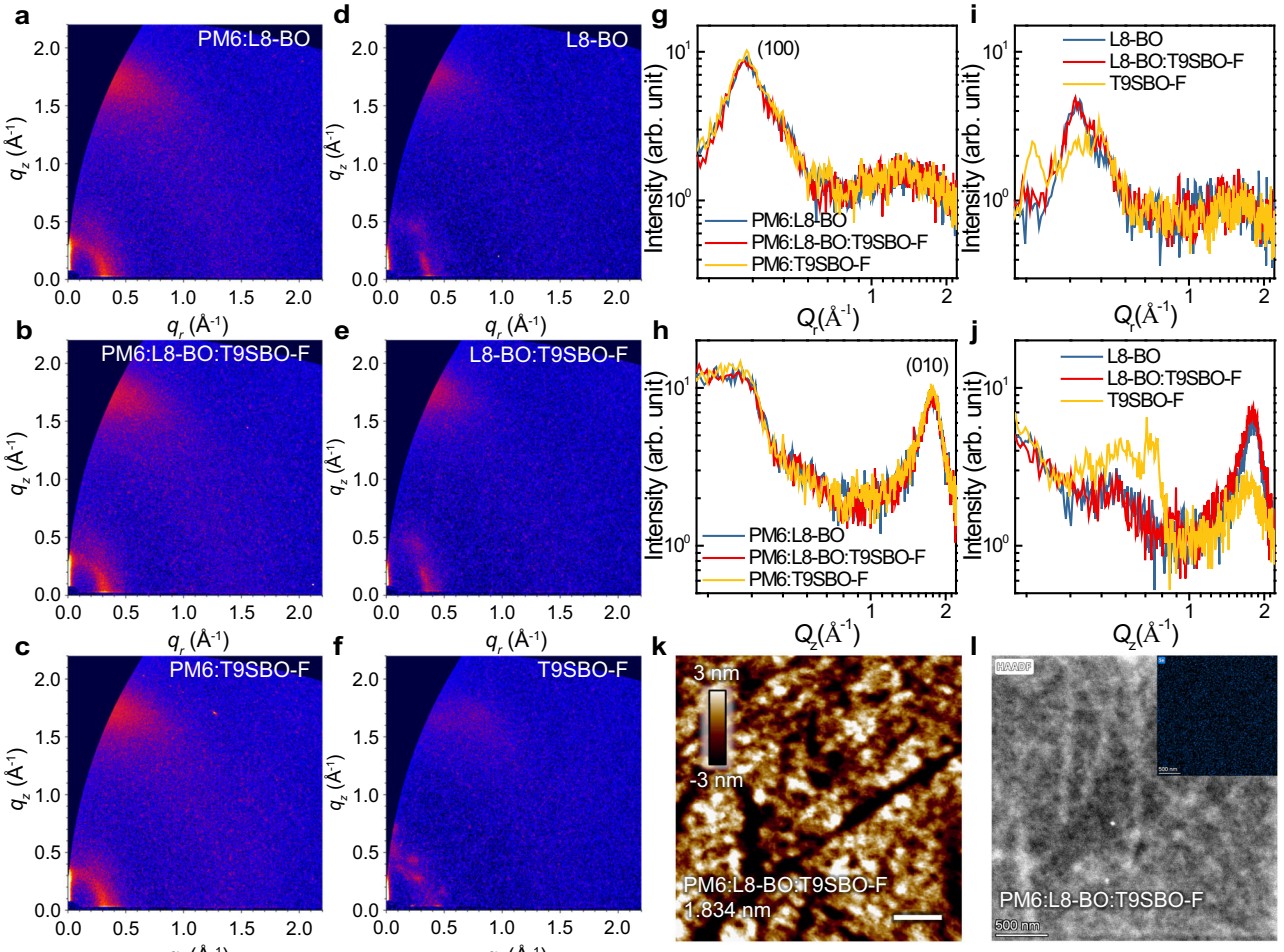

**Fig. 4 | Morphology study of the blend and neat films. a–f** 2D GIWAXS diffraction pattern of blend and neat film. 1D line-cut profiles of **g** blend in-plane direction, **h** blend out-of-plane direction, **i** neat film in-plane direction, and **j** neat film out-of-plane direction. **k** AFM image and **l** HAADF image of PM6:L8-BO:T9SBO-F sample. The inset image of (**l**) shows the EDX mapping result of Se element. Source data are provided as a Source Data file.

refraction data and resolved the π-conjugated part of the molecule. The crystal exhibits four enantiomers, labeled as *M-/M'-* and *P-/P'-* enantiomers (Supplementary Fig. 12a). The π–π stacking distances between different type of enantiomers are 3.37 and 3.32 Å, which are smaller than that of classic NFA, Y6 (3.64 Å)[22]. The disordered stacking of T9SBO-F in both IP and OOP directions can be attributed to the tight molecular packing that results in multiple diffraction peaks in other directions[43]. Note that L8-BO (branching flanking alkyl chain) and T9SBO-F (linear flanking alkyl chain) present different types of 3D packing network, where L8-BO shows closer packing and higher packing density than T9SBO-F. Therefore, L8-BO single crystal shows shorter average π–π stacking distance of 3.19 Å[6], which is consistent with the results retrieved from GIWAXS characterizations.

We then implemented grazing-incidence small-angle scattering (GISAXS) to reveal the nanoscale phase separation in the blend films. Supplementary Fig. 13d exhibits the fitted pure phase domain size of the blend films. The ternary blend shows enlarged domain size of 56 nm compared to ~20 nm of the binary films, correlating to an enhanced phase separation and possible alloy formation, which reduces the encountering chance for the separated charges and thus suppresses their recombination. To supplement the alloy formation, we plotted the $V_{oc}$ dependence on the T9SBO-F ratio in the acceptor (Supplementary Fig. 14), which matches well with the calculated $V_{oc}$ (mismatch <0.15%), supporting possible alloy formation between L8-BO and T9SBO-F[44,45].

## Surface and bulk morphology investigation

We also probe the surface morphology by atomic force microscopy (AFM). As shown in Supplementary Fig. 15, L8-BO and L8-BO:T9SBO-F display a smooth morphology with a root mean square roughness ($R_{RMS}$) of 1.175 and 1.011 nm, respectively. In contrast, the T9SBO-F film shows a granular morphology with more chaotic arrangement, and the phase image shows very sharp contrast, which explains the disordered stacking feature in the GIWAXS. It also shows an increased $R_{RMS}$ of 2.090 nm. After blending the acceptor with polymer PM6, a clear fibril morphology is captured in Supplementary Fig. 16, supporting that the acceptors is well dispersed in the fibril polymer network. Simultaneously, such network also helps to regulate the T9SBO-F stacking. The optimized PM6:L8-BO:T9SBO-F ternary film thus presents $R_{RMS}$ of 1.834 nm.

Except for the surface morphology, we also conducted high-angle annular dark-field scanning transmission electron microscopy (HAADF-STEM) on the blend samples to reveal the bulk morphology (Supplementary Fig. 17). All the blends exhibit fibril morphology with a polymer-interpenetrating network, which is in consistent with the AFM results. Notably, the PM6:T9SBO-F shows a more pronounced granular feature in the bulk region compared to the other two films, which is assigned to the T9SBO-F domains. In addition, the Se-element mapping of PM6:L8-BO:T9SBO-F in Fig. 4l shows a very uniform distribution, suggesting that T9SBO-F is evenly dispersed in the PM6:L8-BO framework. Note that the film morphology is the other factor that

influences the exciton dissociation and hole-transfer time, except for $\varepsilon_r$ discussed in Fig. 2. The similar morphology and molecular stacking of the blend films confirms the reduced hole-transfer time is mainly due to the improved $\varepsilon_r$.

Combining the morphological characterizations, it is likely that T9SBO-F tends to be more disordered stacking in both IP and OOP directions. While mixing it with PM6 helps to regulate its molecular stacking to form a fibril pattern, there is very slight granular morphology in the bulk region that is referred as T9SBO-F domain, which causes the increased recombination loss of PM6:T9SBO-F device and slightly lower performance. In contrast, uniformly dispersing T9SBO-F in PM6:L8-BO framework can effectively interrupt the formation of disordered T9SBO-F aggregation to form alloys with a pronounced face-on orientation, which simultaneously achieves faster exciton/carrier kinetics and suppressed bimolecular recombination.

### Device operational stability
Finally, device stability plays an essential role during operation, particularly there are concerns about the chemical stability of Se compared to S component. To verify the long-term OSC stability w/o Se substitution, we have measured the encapsulated device in ambient condition with maximum power point (MPP) tracking at 1 sun illumination. As shown in Supplementary Fig. 18, the ternary blend device retains 76.8% of the original efficiency after 1000 h operation, while PM6:L8-BO and PM6:T9SBO-F remain 73.6% and 72.1% of original efficiency, respectively. The major loss comes from $J$sc and FF (Supplementary Fig. 18a, c), which is a typical loss due to donor/acceptor phase separation. The improved efficiency retention of the ternary device could be ascribed to the improved domain size.

## Discussion
We reveal that Se substitution on the NFA central core can increase $\varepsilon_r$, which helps to reduce the Coulombic attraction of electron–hole pairs, correlating to an enhanced kinetic energy of the exciton/carrier. The corresponding hole-transfer time of the NFA has significantly reduced from ~10 ps to ~5 ps. Unexpectedly, the PM6:T9SBO-F film also suffers from faster charge recombination after hundreds of picosecond upon excitation. We unveil this is because T9SBO-F domain has relatively disordered stacking, which increase the electron and hole encountering chance and thus recombination rates. From single-crystal information, the disordered stacking of T9SBO-F domain may originate from the tight molecular packing that results in multiple diffraction peaks in other directions. Inspiringly, blending T9SBO-F into the optimized PM6:L8-BO device can break the disordered aggregation of T9SBO-F to form alloys with ordered orientation and enlarged domain size while still remain improved $\varepsilon_r$. As a result, the optimized film exhibits accelerated hole transfer as well as suppressed bimolecular recombination. The PM6:L8-BO:T9SBO-F device shows a boosted PCE of 19.0% with reduced non-radiative recombination of 0.221 eV and better long-term operation stability. Our initial effort to understand the heteroatom substitution induced $\varepsilon_r$ adjustment, and its correlation with electrical and morphological properties, contributes to material/device design and will further boost the OSC performance.

## Methods
### Chemicals
Poly(3,4-ethylenedioxythiophene):polystyrene sulfonate (PEDOT:PSS, Baytron Al4083) is ordered from H.C. Starck GmbH, Germany. Poly[(2,6-(4,8-bis(5-(2-ethylhexyl-3-fluoro)thiophen-2-yl)-benzo[1,2-b:4,5-b']dithiophene))-alt-(5,5-(1',3'-di-2-thienyl-5',7'-bis(2-ethylhexyl)benzo[1',2'-c:4',5'-c']dithiophene-4,8-dione)]     (PM6),    2,2'-((2Z,2'Z)-((12,13-bis(2-ethylhexyl)−3,9-(2-butyloctyl)−12,13-dihydro-[1,2,5]thiadiazolo[3,4-e]thieno[2'',3'':4',5']thieno[2',3':4,5]pyrrolo[3,2-g]thieno[2',3':4,5]thieno[3,2-b]indole-2,10-diyl)bis(methanylylidene))bis(5,6-difluoro-3-oxo-2,3-dihydro-1H-indene-2,1-diylidene))

dimalononitrile (L8-BO) and Poly[[2,7-bis(2-ethylhexyl)-1,2,3,6,7,8-hexahydro-1,3,6,8-tetraoxobenzo[lmn][3,8]phenanthroline-4,9-diyl]-2,5-thiophenediyl[9,9-bis[3-(dimethylamino)propyl]-9H-fluorene-2,7-diyl]-2,5-thiophenediyl] (PNDIT-F3N) are bought from Solarmer Co., Ltd. Chloroform (GR) were purchased from Sigma-Aldrich. 1,2-dibromobenzene (98%) is obtained from Shanghai Macklin Biochemical Co., Ltd.

### Device fabrication
glass/indium tin oxide (ITO) substrates were cleaned by water, acetone, and ethanol in an ultrasonic bath. UVO treatment is performed on the clean substrates for 20 min. Following that, PEDOT:PSS solution is spin-coated at 4000 revolutions per minute (rpm) on the substrate and the substrates were annealed at 120 °C for 20 min to obtain a PEDOT:PSS thin film (~30 nm). Active layer solution (donor:acceptor ratio of 1:1.2 with a total mass concentration of 16.3 mg/mL in chloroform, 0.5 vol% 1,2-dibromobenzene is added in the solution to regulate the phase separation[46]) is spin-coated on PEDOT:PSS at 3000 rpm and then annealed at 80 °C for 50 min to form a layer thickness of about 100 nm. Next, PNDIT-F3N (1 mg/mL in ethanol with 0.5 vol% acetic acid) is deposited on active layers. The device is finished by thermal evaporation a 100-nm Ag contact electrode. The device structure is: ITO/PEDOT:PSS/Active layer/PNDIT-F3N/Ag.

### General characterizations
UV−Vis absorption spectroscopy was performed on a Shimadzu UV−Visible Spectrophotometer (UV-2600). $J$–$V$ curves were characterized by a Keithley 2400 sourcemeter under 100 mW/cm2 (SS-F5-3 A, Enli Technology Co., Ltd.). The light intensity is calibrated by a standard silicon solar cell (SRC-2020, Enli Technology Co., Ltd.). EQE was probed employing solar cell spectral-response system (QE-R, Enli Technology Co., Ltd.). PL and Picosecond TRPL data were collected by excitation with a femtosecond laser (Coherent Mira900) at 780 nm with a repetition rate of 76 MHz. The beam was focused on encapsulated samples. TPC and TPV were characterized with a 532 pulse laser (130 μJ per pulse and 50 Hz), and the signal is recorded by a 4 GHz Keysight MSO9404A digital oscilloscope. TPV operated at open-circuit condition in series with a 5.5 MΩ resistor and TPC was at short-circuit condition in series with a 50 Ω resistor. GIWAXS and GIWSAXS results were obtained by a Xeuss 2.0 SAXS/WAXS laboratory beamline S3 employing a Cu X-ray source (8.05 keV, 1.54 Å) and a Pilatus3R 300 K detector with an incidence angle of 0.2°, the GIWAXS and GISAXS images in this work are displayed as obtained without further processing. Other morphologies information were characterized by AFM (NT-MDT NTEGRA) and TEM (Thermo Scientific Talos F200X STEM), the TEM images in this work are displayed as obtained without further processing.

### Reporting summary
Further information on research design is available in the Nature Portfolio Reporting Summary linked to this article.

## Data availability
The authors declare all data generated in this study are provided in the Supplementary Information/Source Data file. Crystallographic data for the structures reported in this Article have been deposited at the Cambridge Crystallographic Data Centre (CCDC), under deposition number CCDC 2327063. Copies of the data can be obtained free of charge via https://www.ccdc.cam.ac.uk/structures/. Source data are provided with this paper.

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

## Acknowledgements

W.C.H.C. thanks the Research Grant Council of Hong Kong for the General Research Fund (Grant Nos. 17211220, 17200021, and 17200823), Collaboration Research Fund (C7035-20G), Innovation and Technology Commission of Hong Kong for the Innovation and Technology Fund (ITS/277/21FP), and University Grant Council of the University of Hong Kong for the seed fund (Grant Nos. 202011159254 and 202111159113). A.K.-Y.J. thanks the Innovation and Technology Commission of Hong Kong for the Innovation and Technology Fund (MRP/040/21X). K.S.W. thanks the Research Grant Council of Hong Kong for the Collaboration Research Fund (C6013-19G).

## Author contributions

W.C.H.C. supervised the execution of the project. X.H. and W.C.H.C. conceived this project and designed the experiments. X.H. fabricated and characterized the photovoltaic cells. F.Q. and A.K.-Y.J. provided T9SBO-F NFA and offered suggestions about device optimization and characterizations. X.Z. and K.S.W. measured and analyzed the TA data. Y.L. performed the EL and FTPS-EQE measurements. H.L. and X.L carried out the GIWAXS measurement. X.H. and W.C.H.C. drafted the manuscript. All authors participated in the discussions for manuscript preparation. We thank you for the technical discussion with Francis Lin from the City University of Hong Kong.

## Competing interests

The authors declare no competing interests.
