## [Peer Review File · Nature Communications]

Selenium Substitution for Dielectric Constant Improvement and Hole-transfer Acceleration in Non-fullerene Organic Solar CellsReviewers' Comments:

Reviewer #1:

Remarks to the Author:

Comments

The authors report that selenium (Se)-substituted non-fullerene acceptor (Se-NFA, named as T9SBO-F in this manuscript) has a high dielectric constant and shows improved fibril morphology when blending with PM6:LB-BO as a ternary film. Therefore, they achieved a power conversion efficiency (PCE) of ~19.0% owing to enhanced exciton dissociation and reduced non-radiative recombination loss, which were confirmed by various chemical, structural, optical, and photophysical studies. Photovoltaic absorbers with a high dielectric constant should be used to easily separate the photogenerated excitons and improve device performance; however, a dielectric constant value of Se-NFA is still much lower than conventional inorganic semiconductors and they should show more results of electrical properties such as dielectric constant in binary and/or ternary films. Although improving efficiency is important, what is more important in the OPV research part is device stability in air and/or real operating conditions. But, there are no studies and results associated with device stability in this manuscript. I am now thinking that this manuscript is overall well-written and well-organized but can be published in Nature Communications after revision. Detailed comments are provided as below.

1. In terms of lower electronegativity and larger atomic radius, they claimed that Se-NFA shows higher dielectric constant and device performance than conventional S-based NFA. I am wondering about the use of Tellurium (Te) showing instead of S and Se. Is it possible to synthesize Te-based NFA? If it is possible, how do you expect the electrical properties and device performance?

2. Se has a lower reduction potential than S. This means that Se can be more easily oxidized and thus the OPV device composed of Se-NFA might show lower device stability compared to that composed of conventional S-based NFA. They should show the device stability results evaluated in the air and/or real operating conditions (with continuous 1 sun).

3. It is expected that increasing the dielectric constant of NFA can also increase the diffusion distance of absorbers. In this study, it is necessary to confirm electrical property and device performance according to changes in the thickness of the binary or ternary film as well as the contents of Se-NFA, LB-BO, and PM6.

Reviewer #2:

Remarks to the Author:

The authors replaced the outmost thiophene rings of non-fullerene acceptor (NFA) L8-BO with selenophen rings and synthesized a NFA named T9SBO-F. They found the Se substitution can increase the ϵ_r from 3.96 to 5.04 and reduce the hole transfer time from ~10 ps to ~5 ps in the corresponding blend film with PM6. In addition, they unveiled that the faster charge recombination in PM6: T9SBO-F film can be attributed to the more disordered stacking of T9SBO-F domain, which may originate from the bigger Se atomic radius compared to S. Thanks to the more ordered aggregation of T9SBO-F and improved ϵ_r in PM6:L8-BO:T9SBO-F film, the ternary-blend device exhibited a higher PCE of 19.0% relative to the binary-blend devices.

However, on one hand, the Se substitution has been widely used in NFA field and the molecular design in this manuscript is not novel enough. Very similar NFAs with two Se substitution has been reported in previous literatures. e.g., (1) J. Am. Chem. Soc. 2020, 142, 15246. (2) ACS Energy Lett. 2020, 5, 3415. (3) ACS Energy Lett. 2021, 6, 9. (4) Angew. Chem. Int. Ed. 2021, 60, 19241. (5) Angew. Chem. Int. Ed. 2022, 134, e202206930. (6) Mater. Horiz. 2022, 9, 403.

On the other hand, some results and explanations are questionable.

(1) The exciton dissociation happens at the pico- to nano-second timescale. Therefore, the higher ϵ_r at 103-105 Hz is not suitable to explain the reduced hole transfer time.

(2) The hole transfer time τ_2 (~ 10 ps for PM6:L8-BO and ~ 5 ps for PM6:T9SBO-F) is mainly due to the diffusion of the exciton/hole in the bulk acceptor domains and the authors attribute the reduced τ_2 to the increased ϵ_r . How do the ϵ_r influence the diffusion process?

(3) The authors mentioned that "larger atom radius enables Se a more polarizable surface, which contributes to stronger intermolecular interactions of the NFAs" and "T9SBO-F tends to be more disordered stacking in both IP and OOP directions due to the bigger Se atomic radius", which seems a little contradictory. The longer out-of-plane π - π stacking distance of T9SBO-F is a little odd and the single-crystal data should be supplied to analyze this phenomenon sufficiently.

(4) Some literatures (e.g. J. Phys. Chem. Lett. 2016, 7, 4495–4500) have pointed out that the disorder can also be beneficial to the exciton dissociation. How can the authors exclude the influence of chaotic granular T9SBO-F domain in blend films on the reduced hole transfer time.

(5) The difference between JSC from J-V curve and JSC from EQE measurement is larger than 5%, and the PCE should be certified. In addition, the authors claimed that "such disordered packing mode of T9SBO-F molecule will limit charge transport". Hence, the mobility is supposed to be measured and compared.

Therefore, I am not able to suggest its publication in Nature Communications.

Reviewer #3:

Remarks to the Author:

This is an interesting study that involves substituting two sulfur atoms on the central core of L8-BO small molecule acceptor with two selenium atoms to form T9SBO-F such that the molecule dielectric constant (ϵ_r) value of T9SBO-F is larger than that of L8-BO (ϵ_r 5.0 vs. 4.0), without significantly altering the molecular geometry. From the transient absorption (TA) analyses, the PM6:T9SBO film shows faster hole transfer of ~ 5 ps as compared to that of the PM6:L8-BO blend film (~ 10 ps). PM6:L8-BO: T9SBO-F ternary blend devices show champion power conversion efficiency (PCE) of 19%, as compared to 18.5% and 18.4% for PM6:L8-BO and PM6: T9SBO-F binary blend devices, respectively.

There are a few points that the authors must address prior to acceptance of this manuscript for publication.

First, the authors focus mostly on the transient absorption (TA) analyses that show dynamics data of PM6:T9SBO-F with a τ_1 of 0.49 ps and τ_2 of 4.83 ps and PM6:L8-BO with τ_1 of 1.18 ps and τ_2 of 9.15 ps, indicating enhanced hole-transfer in the PM6:T9SBO-F system and the ternary film also presents faster hole-transfer with τ_1 of 0.89 ps and τ_2 of 6.21 ps than PM6:L8-BO.

But, by looking into the PM6:L8-BO: T9SBO-F ternary blend device photovoltaic parameters, the enhancement in PCE mostly resulted from a combination of higher Voc value, as compared to that of PM6: T9SBO-F binary blend device, and higher Jsc value, as compared to that of PM6:L8-BO binary blend device, along with decent fill factor. The authors need to address the link between the photovoltaic parameters and TA results in details.

Second, for the ternary blend that incorporates one polymer donor and two small molecule acceptors, as in this case, that have similar chemical structures, is there any formation of alloy acceptors, as demonstrated in the paper "The Journal of Materials Chemistry A, 2022, 10 (43), 23037-23046?" A rough calculation shows that in a binary alloy acceptor system when the ratio of L8-BO: T9SBO-F is 1:0.2 will give a Voc value of 0.876 V that is close to the measured Voc value of 0.881V for the PM6:L8-BO: T9SBO-F ternary blend device. The 1-D WAXS curves also show similar pack features in the PM6:L8-BO and PM6:T9SBO-F systems in figure 4 g and i.

Third, from supplementary information, Figure 3b, the UV-Vis spectra show extended light absorption for the case of T9SBO-F film, is this also the reason that the EQE value is slightly larger in Figure 1?

Fourth, Figure 1 should be revised to clearly demonstrate L8-BO and T9SBO-F molecular structures instead of just the core structures.

Reviewer #1:

The authors report that selenium (Se)-substituted non-fullerene acceptor (Se-NFA, named as T9SBO-F in this manuscript) has a high dielectric constant and shows improved fibril morphology when blending with PM6:LB-BO as a ternary film. Therefore, they achieved a power conversion efficiency (PCE) of ~19.0% owing to enhanced exciton dissociation and reduced non-radiative recombination loss, which were confirmed by various chemical, structural, optical, and photophysical studies. Photovoltaic absorbers with a high dielectric constant should be used to easily separate the photogenerated excitons and improve device performance; however, a dielectric constant value of Se-NFA is still much lower than conventional inorganic semiconductors and they should show more results of electrical properties such as dielectric constant in binary and/or ternary films. Although improving efficiency is important, what is more important in the OPV research part is device stability in air and/or real operating conditions. But, there are no studies and results associated with device stability in this manuscript. I am now thinking that this manuscript is overall well-written and well-organized but can be published in Nature Communications after revision. Detailed comments are provided as below.

1. In terms of lower electronegativity and larger atomic radius, they claimed that Se-NFA shows higher dielectric constant and device performance than conventional S-based NFA. I am wondering about the use of Tellurium (Te) showing instead of S and Se. Is it possible to synthesize Te-based NFA? If it is possible, how do you expect the electrical properties and device performance?

Response: Thank you for the good comment. We are aware that Tellurium (Te) may introduce higher polarizability which further improves the dielectric constant of corresponding materials. However, it is practically difficult to synthesize Te-based NFAs because of the barely controllable reactivity and high toxicity. For example, the C–Te bond in tellurophene can be easily cleaved by catalysts such as Pt(II) and Pd(II), reagents such as BuLi, and other strong bases/nucleophiles, due to its lower bonding energy (200 kJ mol^{-1}) in comparison with C–S (272 kJ mol^{-1}) and C–Se bonds (234 kJ mol^{-1}).^[1-4] Moreover, the strong coordination strength of Te will likely poison the transition metal-based catalysts during coupling reactions, rendering the synthesis unfeasible.^[5]

2. Se has a lower reduction potential than S. This means that Se can be more easily oxidized and thus the OPV device composed of Se-NFA might show lower device stability compared to

that composed of conventional S-based NFA. They should show the device stability results evaluated in the air and/or real operating conditions (with continuous 1 sun).

Response: Thank you for the thoughtful comment. We agree that Se could be more easily oxidized. However, currently, the degradation of OSC devices is mainly induced by the quasi-stable phase separation in the organic BHJ blend.^[6] The meta-stable donor/acceptor will further phase separate during the device operation. To verify the OSC stability w/o Se substitution, we have measured the encapsulated device stability in ambient condition with maximum power point (MPP) tracking at 1 sun illumination. As shown in Figure R1, PM6:L8-BO:T9SBO-F ternary device retains 76.8% of the original efficiency after 1000 h operation, while PM6:L8-BO and PM6:T9SBO-F remain 73.6% and 72.1% of original efficiency, respectively. The major loss comes from J_{sc} and FF (Figure R1a and R1c), which is a typical loss due to phase separation. The improved efficiency retention of the ternary device is ascribed to the improved domain size. As shown in Figure R2, we implemented grazing-incidence small-angle scattering (GISAXS) to reveal the nanoscale phase separation in the films. Figure R2d exhibits the fitted pure phase domain size of the blend films. The ternary blend shows enlarged domain size of 56 nm compared to ~20 nm of the binary films, correlating to an enhanced phase separation and the better J_{sc} and FF retention.

Figure R1. Stability. (a) J_{sc} (b) V_{oc} (c) FF and (d) PCE of the encapsulated devices in ambient condition with MPP tracking and continuous LED light illumination (100 mW/cm^2).

Figure R2. Nanoscale phase separation. 2D GISAXS pattern of (a) PM6:L8-BO (b) PM6:L8-BO:T9SBO-F and (c) PM6:T9SBO-F film. (d) The 1D line-cut in-plane GISAXS profile and the fitted results.

3. It is expected that increasing the dielectric constant of NFA can also increase the diffusion distance of absorbers. In this study, it is necessary to confirm electrical property and device performance according to changes in the thickness of the binary or ternary film as well as the contents of Se-NFA, LB-BO, and PM6.

Response: Thank you for the constructive comment. We have fabricated devices with different L8-BO:T9SBO-F ratios covering PM6:L8-BO:T9SBO-F: 1:1.2:0 (0%), 1:1:0.2 (16.7%), 1:0.6:0.6 (50%), 1:0.2:1 (83.3%), 1:0:1.2 (100%). The detailed device performances are concluded in Table R1. Figure R3a and R3b show the J - V curve and EQE spectra of different devices. With the increase of T9SBO-F ratio in the acceptor, the device V_{oc} gradually decreases from 0.888 V (0%), to 0.881 V (16.7%), 0.858 V (50%), 0.837 V (83.3%) and 0.829 V (100%). While increasing the T9SBO-F ratio helps to extend the absorption to near infrared region (Figure R2b), the EQE value keeps decreasing in the range of 650-800 nm, which is ascribed to the increased charge recombination. Overall, the J_{sc} gradually increases from 26.0

mA/cm² (0%), to 27.1 mA/cm² (16.7%), 27.2 mA/cm² (50%), 27.8 mA/cm² (83.3%) and 28.1 mA/cm² (100%). When blending PM6:L8-BO:T9SBO-F at a ratio of 1:1:0.2, the device presents the highest PCE of 19.0%.

We then measured the dielectric constant (ϵ_r) of different donor/acceptor blends. With the increase of T9SBO-F ratio, ϵ_r increases from 3.36 (0%), to 3.51 (16.7%), 3.56 (50%), 3.93 (83.3%) and 4.15 (100%). We further measured the hole (μ_h) and electron (μ_e) mobility of the active layer with different amount of T9SBO-F based on space-charge-limited current (SCLC) method. The detailed measurement and device configuration are described in experimental sections in Supplementary information. As shown in Figure R3d, R3e and Table R1, all the blend shows similar μ_h of around $10.5 \times 10^{-4} \text{ cm}^2 \text{ V}^{-1} \text{ s}^{-1}$, which coincides with the similar fibril PM6 morphology of the active layers. For μ_e , the blends with different T9SBO-F ratios did not show significant difference. Simultaneously, the average series resistance (R_s) of different devices are around 65 Ω with the PM6:T9SBO-F presents the lowest R_s of 63.4 Ω .

To further demonstrate the electrical properties of the Se-NFA, we followed the reviewer's suggestions and fabricated thick-layer (~300 nm) OSC based on the binary and best-performed ternary (1:1:0.2) devices, the photovoltaic performance and electrical properties are concluded in Figure R4 and Table R2. Compared to thin OSC (100 nm), the thick OSC shows improved J_{sc} , slightly lower V_{oc} and significantly reduced FF. Among them, the ternary presents the best PCE of 16% with a J_{sc} of 27.7 mA/cm², V_{oc} of 0.860 V and FF of 66.9%. We also measured the μ_h and μ_e of the thick devices. While all the devices exhibit similar μ_h around $8 \times 10^{-4} \text{ cm}^2 \text{ V}^{-1} \text{ s}^{-1}$, the PM6:T9SBO-F shows a μ_e of $40.7 \times 10^{-4} \text{ cm}^2 \text{ V}^{-1} \text{ s}^{-1}$, which is obviously higher than $12.4 \times 10^{-4} \text{ cm}^2 \text{ V}^{-1} \text{ s}^{-1}$ of PM6:L8-BO and $13.6 \times 10^{-4} \text{ cm}^2 \text{ V}^{-1} \text{ s}^{-1}$ of PM6:L8-BO:T9SBO-F. This can be ascribed to the reduced R_s (Figure R4e), which is beneficial for carrier transport. The results support that NFA with a high ϵ_r shows electron mobility improvement particularly in thick devices. However, the significantly unbalanced μ_h/μ_e (0.17) of the PM6:T9SBO-F has limited the device performance, as confirmed by the reduced shunt resistance (R_{sh} , Figure R4f), leading to severe charge recombination and thus poor FF.

Figure R3. Device performance and electrical properties of OSC with different Se-NFA ratios. (a) J - V curve, (b) EQE spectra, (c) dielectric constant, (d) SCLC hole-only device, (e) SCLC electron-only device and (f) device series resistance. The ratio shown in the figures are weight ratio of PM6:L8-BO:T9SBO-F.

Figure R4. Device performance and electrical properties of thick OSC (active layer ~300 nm). (a) J - V curve, (b) EQE spectra, (c) SCLC hole-only device, (d) SCLC electron-only device, (e) device series resistance and (f) device shunt resistance. The ratio of PM6:L8-BO:T9SBO-F thick device is optimized at 1:1:0.2.

References

1. Pacholska-Dudziak, E.; Vetter, G.; Góratowska, A.; Białońska, A.; Latos-Grażyński, L., Chemistry inside a Porphyrin Skeleton: Platinacyclopentadiene from Tellurophene. *Chemistry – A European Journal* **2020**, *26* (68), 16011-16018.

2. Pacholska-Dudziak, E.; Szczepaniak, M.; Książek, A.; Latos-Grażyński, L., A Porphyrin Skeleton Containing a Palladacyclopentadiene. *Angewandte Chemie International Edition* **2013**, *52* (34), 8898-8903.
3. Zheng, F.; Tan, S.-E.; Yanamoto, Y.; Shida, N.; Nishiyama, H.; Inagi, S.; Tomita, I., Preparation of a germole-containing π -conjugated polymer by the Te–Li exchange reaction of a tellurophene-containing polymer. *NPG Asia Materials* **2020**, *12* (1), 41.
4. Chivers, T.; Laitinen, R. S., Tellurium: a maverick among the chalcogens. *Chemical Society Reviews* **2015**, *44* (7), 1725-1739.
5. Hasegawa, M.; Haga, S.; Nishinaga, T.; Mazaki, Y., Selenacalix[4]selenophene: Synthesis, Structure, and Gel Formation of Cyclic Selenoether of Selenophene. *Organic Letters* **2020**, *22* (10), 3755-3758.
6. Liang, Y.; Zhang, D.; Wu, Z.; Jia, T.; Lüer, L.; Tang, H.; Hong, L.; Zhang, J.; Zhang, K.; Brabec, C. J.; Li, N.; Huang, F., Organic solar cells using oligomer acceptors for improved stability and efficiency. *Nature Energy* **2022**, *7* (12), 1180-1190.

Table R1. Device performance and electrical properties of OSC with different Se-NFA ratios (active layer ~100 nm).

PM6:L8-BO:T9SBO-F	ϵ_r	J_{SC} (mA/cm ²)	$J_{SC, EQE}$ (mA/cm ²)	V_{oc} (V)	FF (%)	PCE (%)	μ_h (cm ² V ⁻¹ s ⁻¹)	μ_e (cm ² V ⁻¹ s ⁻¹)	μ_h/μ_e	R_s (Ω)	R_{sh} (Ω)
1:1.2:0	3.36	26.0 (25.9 ± 0.2)	24.8	0.888 (0.888 ± 0.002)	80.4 (79.9 ± 0.4)	18.5 (18.3 ± 0.1)	10.37	10.69	0.97	70.2	50745
1:1:0.2	3.51	27.1 (26.8 ± 0.2)	25.6	0.881 (0.877 ± 0.002)	79.6 (79.8 ± 0.4)	19.0 (18.8 ± 0.1)	10.07	10.87	0.93	65.8	71827
1:0.6:0.6	3.56	27.2 (27.0 ± 0.2)	25.7	0.858 (0.858 ± 0.002)	79.1 (78.9 ± 0.2)	18.5 (18.3 ± 0.1)	10.74	10.12	1.06	69.3	51901
1:0.2:1	3.93	27.8 (27.7 ± 0.1)	26.1	0.837 (0.836 ± 0.002)	79.0 (78.5 ± 0.3)	18.4 (18.2 ± 0.1)	10.27	9.66	1.06	66.2	58055
1:0:1.2	4.15	28.1 (27.9 ± 0.2)	26.4	0.829 (0.827 ± 0.002)	78.9 (78.9 ± 0.2)	18.4 (18.2 ± 0.1)	10.78	9.97	1.08	63.4	52945

Table R2. Device performance and electrical properties of thick OSC (active layer ~300 nm).

	Thickness (nm)	J_{SC} (mA/cm ²)	$J_{SC, EQE}$ (mA/cm ²)	V_{oc} (V)	FF (%)	PCE (%)	μ_h (cm ² V ⁻¹ s ⁻¹)	μ_e (cm ² V ⁻¹ s ⁻¹)	μ_h/μ_e	R_s (Ω)	R_{sh} (Ω)
PM6:L8-BO	295 ± 7	26.0 (25.8 ± 0.2)	25.7	0.874 (0.878 ± 0.003)	69.0 (68.2 ± 0.4)	15.7 (15.4 ± 0.1)	8.37	12.44	0.67	94.1	32197
PM6:L8-BO:T9SBO-F	298 ± 5	27.7 (27.4 ± 0.3)	26.3	0.860 (0.861 ± 0.003)	66.9 (66.6 ± 0.5)	16.0 (15.7 ± 0.2)	8.69	13.63	0.64	93.2	26002
PM6:T9SBO-F	293 ± 6	29.1 (29.0 ± 0.2)	27.4	0.816 (0.816 ± 0.001)	62.5 (62.1 ± 0.5)	14.9 (14.7 ± 0.1)	7.12	40.71	0.17	87.9	15970

Reviewer #2:

The authors replaced the outmost thiophene rings of non-fullerene acceptor (NFA) L8-BO with selenophene rings and synthesized a NFA named T9SBO-F. They found the Se substitution can increase the ϵ_r from 3.96 to 5.04 and reduce the hole transfer time from ~10 ps to ~5 ps in the corresponding blend film with PM6. In addition, they unveiled that the faster charge recombination in PM6: T9SBO-F film can be attributed to the more disordered stacking of T9SBO-F domain, which may originate from the bigger Se atomic radius compared to S. Thanks to the more ordered aggregation of T9SBO-F and improved ϵ_r in PM6:L8-BO:T9SBO-F film, the ternary-blend device exhibited a higher PCE of 19.0% relative to the binary-blend devices.

However, on one hand, the Se substitution has been widely used in NFA field and the molecular design in this manuscript is not novel enough. Very similar NFAs with two Se substitution has been reported in previous literatures. e.g., (1) J. Am. Chem. Soc. 2020, 142, 15246. (2) ACS Energy Lett. 2020, 5, 3415. (3) ACS Energy Lett. 2021, 6, 9. (4) Angew. Chem. Int. Ed. 2021, 60, 19241. (5) Angew. Chem. Int. Ed. 2022, 134, e202206930. (6) Mater. Horiz. 2022, 9, 403. On the other hand, some results and explanations are questionable. Therefore, I am not able to suggest its publication in Nature Communications.

Response: Thank you for reviewing the manuscript. It is a fact that Se substitution has been widely reported and employed in NFAs, and we have no intention to claim the chemical structure and Se substitution in this work is unique. In contrast, it is because of the wide reports of Se substitution and their excellent device performance that triggers our interest to reveal the reasons behind and to investigate the relationship between material properties, film morphology and exciton/carrier kinetics. In organic solar cells, a unique intermediate state (e.g. charge-transfer state) exists to mediate the local exciton state and charge separated state mainly because the high electron-hole binding energy in OSC requires excess energy for exciton dissociation other than excitation energy. However, the existence of intermediate state enables another charge recombination channel and increases the total energy loss. One of the fundamental strategies to boost OSC performance is to improve the material dielectric constant (ϵ_r) and thus to facilitate the exciton dissociation and charge separation. As a “heavy” atom compared to S, Se substitution to S brings the promise to improve ϵ_r .

In this work, we firstly compared Se substitution on different positions and reveals that double Se substitution features the best dipole moment improvement. Following that, we probed the

ϵ_r of the NFAs and measured the corresponding device performance. With the revised manuscript based on the reviewers' comments, we systematically analysed the exciton dissociation and charge recombination dynamics, quantified the related energy loss, and revealed their relationship with the molecular stacking and film morphology. We show that the morphology and PCE of PM6:L8-BO and PM6:T9SBO-F blend are very close but their exciton/charge dynamics show significant difference, which highlights the important role played by ϵ_r . Then, we conclude that the improved ϵ_r of NFA can undoubtedly fasten the exciton dissociation and hole-transfer kinetics in the blend. However, the film morphology will also influence the exciton/charge kinetics, which has been ameliorated to suppress the bimolecular recombination. The work reveals the Se-substitution induced ϵ_r adjustment and its relationship with the exciton/charge kinetics, energy loss and morphology, which offers guideline for proper material/film design and boosts device performance.

The point-by-point response is attached below to address the reviewer's comments.

(1) The exciton dissociation happens at the pico- to nano-second timescale. Therefore, the higher ϵ_r at 10^3 - 10^5 Hz is not suitable to explain the reduced hole transfer time.

Response: Thank you for the detailed reviewing. The OSC devices can be considered as an equivalent circuit shown in the inset image of Figure R5. The fitted results gives a capacitance of 2.4 nF, R_s of 63.1 Ω and R_p of 1.9 M Ω , from which we are able to obtain the effective resistor-capacitor (RC) characteristic time (τ) of ~0.2 μ s. Next, we can calculate the cut-off frequency (f_c) of the equivalent circuit by the equation:

$$f_c = \frac{1}{2\pi\tau}$$

An f_c of 8×10^5 Hz is obtained, which defines the upper limit of the AC frequency regime for the dielectric constant (ϵ_r) measurement.^[1]

In addition, there are reports concluded the dielectric response modes at different AC frequencies (e.g. optical dielectric response at $\sim 10^{14}$ Hz from electron density; ionic dielectric response at $\sim 10^{12}$ Hz from lattice vibration; dipolar dielectric response at $\sim 10^9$ Hz from dipolar species; and space-charge dielectric response at $< 10^6$ Hz from both electronic or ionic space charge).^[2] The ϵ_r at 10^3 - 10^5 Hz describes the space-charge dielectric response of the material from the delocalized space charges.

Moreover, in this frequency region, the active layer will be fully depleted, which can be employed to indicate the ϵ_r value of the studied materials.^[3] This AC frequency region (10^3 - 10^5 Hz) has been widely used to show the ϵ_r of NFA OPV^[4-6] and perovskite solar cells^[1, 7]. Based on the above considerations, we believe it would be suitable to use ϵ_r at 10^3 - 10^5 Hz to explain the exciton, delocalized singlet excitons (DSE) and separated charge kinetics in the manuscript.

Figure R5. The impedance spectrum of the PM6:T9SBO-F device. The inset figure shows the equivalent circuit. The measurement is carried out at dark condition with a DC bias of 30 mV.

(2) The hole transfer time τ_2 (~ 10 ps for PM6:L8-BO and ~ 5 ps for PM6:T9SBO-F) is mainly due to the diffusion of the exciton/hole in the bulk acceptor domains and the authors attribute the reduced τ_2 to the increased ϵ_r . How do the ϵ_r influence the diffusion process?

Response: Thank you for the good question. We agree with the reviewer that the hole-transfer time deduction is mainly due to the faster diffusion of the exciton/hole in the bulk acceptor domain. However, it does not counter our conclusion because ϵ_r of the material is a decisive parameter to the exciton/hole kinetics. The kinetic energy of an exciton can be approximated by the following equation:

$$K_E = h\nu - \phi - E_b$$

where K_E is the kinetic energy, $h\nu$ is the excitation energy, ϕ is the energy gap of photo-excitation and transitions between different states (LE state \rightarrow DSE/CT state \rightarrow CS state), and E_b is the electron-hole binding energy. E_b can be approximated by the following equation:^[8-9]

$$E_b \approx e^2 / (4\pi\epsilon_0\epsilon_r R)$$

where e is the elementary charge, ϵ_0 is the vacuum dielectric constant, ϵ_r is the relative dielectric constant and R is the average distance of the electron and hole.

Based on the above relationship, a higher ϵ_r can reduce the exciton binding energy and enable the exciton/carrier with more kinetic energy, and finally facilitates the exciton dissociation and hole transfer. To supplement, we have measured E_b ($E_b = E_g - E_g^{\text{opt}}$) by subtracting the optical gap (E_g^{opt} , absorption onset) from the fundamental bandgap ($E_g = \text{IP} - \text{EA}$) of the NFAs.^[10] Our results show that T9SBO-F features 20 meV less E_b than L8-BO neat film.

(3) The authors mentioned that “larger atom radius enables Se a more polarizable surface, which contributes to stronger intermolecular interactions of the NFAs” and “T9SBO-F tends to be more disordered stacking in both IP and OOP directions due to the bigger Se atomic radius”, which seems a little contradictory. The longer out-of-plane π - π stacking distance of T9SBO-F is a little odd and the single-crystal data should be supplied to analyze this phenomenon sufficiently.

Response: Thank you for the constructive suggestion. We totally agree with the reviewer that single-crystal data may offer helpful information to further understand the microstructures of NFAs in the condensed phase. We did make attempts to grow the single crystal of T9SBO-F and collect the refraction data. We successfully resolved the π -conjugated part of the molecule, and we are confident with the analysis of π - π stacking distance from the single-crystal data as described below. However, there is certain degree of disorder induced by the long, branching alkyl chains on the molecules, of which the intermolecular interactions we intend not to make discussion in this work.

Compared with the single-crystal data of the classic NFA, Y6, shown previously,^[11] T9SBO-F exhibits four enantiomers, labeled as M -/ M' - and P -/ P' -enantiomers (Figure R6). We find that the π - π stacking distances between different type of enantiomers are 3.37 and 3.32 Å, which are smaller than that (3.64 Å) of Y6, indicating the incorporation of Se atom can efficiently enhance the intermolecular interactions. For the disordered stacking of T9SBO-F in both IP and OOP directions, this can be attributed to the tight molecular packing that results in multiple diffraction peaks in other directions. Similar phenomenon based on the Se-substituted non-fullerene acceptors is also observed in the reported literature.^[12]

In terms of the different OOP π - π stacking distances of neat T9SBO-F and L8-BO film, this should stem from the different types of 3D packing network between L8-BO (branching flanking alkyl chain) and T9SBO-F (linear flanking alkyl chain), where L8-BO shows closer packing and higher packing density than Y6 and T9SBO-F. As a result, the average π - π stacking distance of L8-BO is 3.19 Å.^[13] This is consistent with the result of the out-of-plane π - π stacking distance retrieved from the GIWAXS characterization.

Figure R6. Molecular packing information. Molecular packing with four enantiomers in the single crystal of T9SBO-F.

(4) Some literatures (e.g. *J. Phys. Chem. Lett.* 2016, 7, 4495–4500) have pointed out that the disorder can also be beneficial to the exciton dissociation. How can the authors exclude the influence of chaotic granular T9SBO-F domain in blend films on the reduced hole transfer time.

Response: Thank you for the reminder. There might be some misunderstanding regarding this question. Firstly, the hole-transfer time is significantly reduced in the donor/acceptor blend. We show in the manuscript that T9SBO-F neat film shows obvious chaotic peaks compared to other two films. However, after blending it with PM6, the PM6 framework helps to regulate the stacking behavior of T9SBO-F, and PM6:T9SBO-F shows similar GIWAXS pattern (Figure 4a-c), GISAXS pattern (Supplementary Fig. 13) and AFM image with PM6:L8-BO (Supplementary Fig. 16).

Although we agree with the reviewer that disorder or well mixing of the donor and acceptor will contribute to the exciton dissociation, the hole-transfer time is significantly reduced (from ~10 ps to ~5 ps) whereas the film morphology and stacking behavior show very slight

difference. Therefore, we state that the reduced hole-transfer time is mainly due to the improved dielectric constant of the material. We have made revision in the manuscript to acknowledge the disorder contribution to the hole-transfer time reduction.

(5) The difference between J_{sc} from J - V curve and J_{sc} from EQE measurement is larger than 5%, and the PCE should be certified. In addition, the authors claimed that “such disordered packing mode of T9SBO-F molecule will limit charge transport”. Hence, the mobility is supposed to be measured and compared.

Response: Thank you for the good suggestion. We noticed the J_{sc} obtained from J - V curve and from EQE spectrum has a mismatch of 5.5% for the best-performed device. Although we failed to book a certification timeslot due to very large volume of demands, to double confirm the result and to exclude possible machine deviation, we measured the J - V curve and EQE spectra of the devices in the co-authors' lab from different institutes (Alex Jen's lab in City University of Hong Kong and the central facilities of the Chinese University of Hong Kong), where they made many certifications in third parties and the certified reports are very close to their lab measured results. As shown in Figure R7a, the ternary device measured in the City University of Hong Kong presents a PCE of 19.1% with J_{sc} of 27.0 mA/cm², V_{oc} of 0.878 V and FF of 80.5%. The integrated J_{sc} from EQE spectrum is 26.6 mA/cm² (Figure R7b), correlating to a mismatch of 1.5% with the J_{sc} from J - V curve. Similarly, the device measured in the Chinese University of Hong Kong shows a J_{sc} mismatch of 2.2%, confirming the reproducible device performance in different institutes (Figure R7c and R7d).

Figure R7. Device performance measured in different institutes. (a) J - V curve and (b) EQE spectrum measured in the City University of Hong Kong. (c) J - V curve and (d) EQE spectrum measured in the Chinese University of Hong Kong.

Secondly, we have also characterized the mobility of different devices as below. Before presenting the data, we would like to clarify again that T9SBO-F neat film present “such disordered packing mode”, but in PM6:T9SBO-F blend film, the molecular stacking is similar compared to PM6:L8-BO films because PM6 fibril framework helps to regulate the stacking.

As shown in Figure R8a, all the blend shows similar μ_h of around $10.5 \cdot 10^{-4} \text{ cm}^2 \text{ V}^{-1} \text{ s}^{-1}$, which coincides with the similar fibril PM6 morphology of the active layers. For μ_e , the blends did not show significant difference with T9SBO-F presents slightly lower value of $9.97 \cdot 10^{-4} \text{ cm}^2 \text{ V}^{-1} \text{ s}^{-1}$, compared to $10.69 \cdot 10^{-4} \text{ cm}^2 \text{ V}^{-1} \text{ s}^{-1}$ of PM6:L8-BO, and $10.87 \cdot 10^{-4} \text{ cm}^2 \text{ V}^{-1} \text{ s}^{-1}$ of the ternary device. Notably, considering that carrier mobility will be influenced by both film morphology and material properties (e.g. ϵ_r), we have revised the statement to “such disordered packing mode of T9SBO-F molecule is not beneficial for the final device performance” to avoid possible misunderstanding.

Figure R8. The carrier mobility of devices. SCLC (a) hole and (b) electron-only devices. The active layer thickness is ~ 100 nm. The unit of the mobility value is $\text{cm}^2 \text{V}^{-1} \text{s}^{-1}$.

References

1. Lin, Q.; Armin, A.; Nagiri, R. C. R.; Burn, P. L.; Meredith, P., Electro-optics of perovskite solar cells. *Nature Photonics* **2015**, *9* (2), 106-112.
2. Su, R.; Xu, Z.; Wu, J.; Luo, D.; Hu, Q.; Yang, W.; Yang, X.; Zhang, R.; Yu, H.; Russell, T. P.; Gong, Q.; Zhang, W.; Zhu, R., Dielectric screening in perovskite photovoltaics. *Nature Communications* **2021**, *12* (1), 2479.
3. Awni, R. A.; Song, Z.; Chen, C.; Li, C.; Wang, C.; Razooqi, M. A.; Chen, L.; Wang, X.; Ellingson, R. J.; Li, J. V.; Yan, Y., Influence of Charge Transport Layers on Capacitance Measured in Halide Perovskite Solar Cells. *Joule* **2020**, *4* (3), 644-657.
4. Liang, H.; Chen, H.; Wang, P.; Zhu, Y.; Zhang, Y.; Feng, W.; Ma, K.; Lin, Y.; Ma, Z.; Long, G.; Li, C.; Kan, B.; Yao, Z.; Zhang, H.; Wan, X.; Chen, Y., Molecular Packing and Dielectric Property Optimization through Peripheral Halogen Swapping Enables Binary Organic Solar Cells with an Efficiency of 18.77%. *Advanced Functional Materials* **2023**, *33* (31), 2301573.
5. Li, T.; Wang, K.; Cai, G.; Li, Y.; Liu, H.; Jia, Y.; Zhang, Z.; Lu, X.; Yang, Y.; Lin, Y., Asymmetric Glycolated Substitution for Enhanced Permittivity and Ecocompatibility of High-Performance Photovoltaic Electron Acceptor. *JACS Au* **2021**, *1* (10), 1733-1742.
6. Zhang, Y.; He, Y.; Zeng, L.; Lüer, L.; Deng, W.; Chen, Y.; Zhou, J.; Wang, Z.; Brabec, C. J.; Wu, H.; Xie, Z.; Duan, C., Unraveling the Role of Non-Fullerene Acceptor with High Dielectric Constant in Organic Solar Cells. *Small* **2023**, *19* (30), 2302314.

7. Cheng, Q.; Wang, B.; Huang, G.; Li, Y.; Li, X.; Chen, J.; Yue, S.; Li, K.; Zhang, H.; Zhang, Y.; Zhou, H., Impact of Strain Relaxation on 2D Ruddlesden–Popper Perovskite Solar Cells. *Angewandte Chemie International Edition* **2022**, *61* (36), e202208264.
8. Zhu, L.; Yi, Y.; Wei, Z., Exciton Binding Energies of Nonfullerene Small Molecule Acceptors: Implication for Exciton Dissociation Driving Forces in Organic Solar Cells. *The Journal of Physical Chemistry C* **2018**, *122* (39), 22309-22316.
9. Wang, J.; Xue, P.; Jiang, Y.; Huo, Y.; Zhan, X., The principles, design and applications of fused-ring electron acceptors. *Nature Reviews Chemistry* **2022**, *6* (9), 614-634.
10. Qian, Y.; Han, Y.; Zhang, X.; Yang, G.; Zhang, G.; Jiang, H.-L., Computation-based regulation of excitonic effects in donor-acceptor covalent organic frameworks for enhanced photocatalysis. *Nature Communications* **2023**, *14* (1), 3083.
11. Lin, F.; Jiang, K.; Kaminsky, W.; Zhu, Z.; Jen, A. K. Y., A Non-fullerene Acceptor with Enhanced Intermolecular π -Core Interaction for High-Performance Organic Solar Cells. *Journal of the American Chemical Society* **2020**, *142* (36), 15246-15251.
12. Yang, C.; An, Q.; Jiang, M.; Ma, X.; Mahmood, A.; Zhang, H.; Zhao, X.; Zhi, H.-F.; Jee, M. H.; Woo, H. Y., Optimized Crystal Framework by Asymmetric Core Isomerization in Selenium-Substituted Acceptor for Efficient Binary Organic Solar Cells. *Angewandte Chemie* **2023**, e202313016.
13. Li, C.; Zhou, J.; Song, J.; Xu, J.; Zhang, H.; Zhang, X.; Guo, J.; Zhu, L.; Wei, D.; Han, G.; Min, J.; Zhang, Y.; Xie, Z.; Yi, Y.; Yan, H.; Gao, F.; Liu, F.; Sun, Y., Non-fullerene acceptors with branched side chains and improved molecular packing to exceed 18% efficiency in organic solar cells. *Nature Energy* **2021**, *6* (6), 605-613.

Reviewer #3:

This is an interesting study that involves substituting two sulfur atoms on the central core of L8-BO small molecule acceptor with two selenium atoms to form T9SBO-F such that the molecule dielectric constant (ϵ_r) value of T9SBO-F is larger than that of L8-BO (ϵ_r 5.0 vs. 4.0), without significantly altering the molecular geometry. From the transient absorption (TA) analyses, the PM6:T9SBO film shows faster hole transfer of ~ 5 ps as compared to that of the PM6:L8-BO blend film (~ 10 ps). PM6:L8-BO:T9SBO-F ternary blend devices show champion power conversion efficiency (PCE) of 19%, as compared to 18.5% and 18.4% for PM6:L8-BO and PM6:T9SBO-F binary blend devices, respectively.

There are a few points that the authors must address prior to acceptance of this manuscript for publication.

1. First, the authors focus mostly on the transient absorption (TA) analyses that show dynamics data of PM6:T9SBO-F with a τ_1 of 0.49 ps and τ_2 of 4.83 ps and PM6:L8-BO with τ_1 of 1.18 ps and τ_2 of 9.15 ps, indicating enhanced hole-transfer in the PM6:T9SBO-F system and the ternary film also presents faster hole-transfer with τ_1 of 0.89 ps and τ_2 of 6.21 ps than PM6:L8-BO. But, by looking into the PM6:L8-BO:T9SBO-F ternary blend device photovoltaic parameters, the enhancement in PCE mostly resulted from a combination of higher V_{oc} value, as compared to that of PM6:T9SBO-F binary blend device, and higher J_{sc} value, as compared to that of PM6:L8-BO binary blend device, along with decent fill factor. The authors need to address the link between the photovoltaic parameters and TA results in details.

Response: Thank you for the constructive comment. We agree with the reviewer that the PCE enhancement in the ternary device is because of the higher V_{oc} , J_{sc} and decent FF. Simply blending different components may help to extend the absorption of the films, but it is the detailed exciton/carrier dynamics that determines whether the absorbed photons can be efficiently converted and contributes to the final V_{oc} , J_{sc} and FF.

In T9SBO-F system, although it presents the fastest hole-transfer (~ 5 ps), its charge recombination is more severe (Figure 2d-f in the manuscript), leading to a larger V_{loss} and slightly decreased FF, which can also be confirmed by the weaker donor GSB of the sample excited at 800 nm.

After dispersing T9SBO-F in PM6:L8-BO, the TA kinetics in Figure 2 exhibits accelerated hole-transfer (~ 7 ps) compared to PM6:L8-BO binary system and simultaneously suppressed

recombination. The delocalized state probed at 1300-1500 nm presents a significantly delayed decay (by ~ 200 ps) in the later charge separated state (after 100 ps delay time), indicating a suppressed bimolecular recombination (Figure 2d in the manuscript). Such reduced recombination is the main reason for the reduced non-radiative recombination loss, contributing to the maintained high V_{oc} in ternary device. Additionally, the reduced recombination also helps to retain the high quantum efficiency in 650-800 nm in the EQE spectra (Figure 1f in the manuscript). Therefore, the ternary OSC presents faster hole-transfer together with suppressed recombination, which simultaneously achieves improved J_{sc} , V_{oc} and decent FF.

To supplement the reasons for the suppressed bimolecular recombination in the ternary device, we have measured GISAXS. Figure R9d exhibits the fitted pure phase domain size of the blend films. The ternary blend shows enlarged domain size of 56 nm compared to ~ 20 nm of the binary films, which reduces the encountering chance for the separated charges and thus suppress their recombination.

Figure R9. Nanoscale phase separation. 2D GISAXS pattern of (a) PM6:L8-BO (b) PM6:L8-BO:T9SBO-F and (c) PM6:T9SBO-F film. (d) The 1D line-cut in-plane GISAXS profile and the fitted results.

2. Second, for the ternary blend that incorporates one polymer donor and two small molecule acceptors, as in this case, that have similar chemical structures, is there any formation of alloy acceptors, as demonstrated in the paper “*The Journal of Materials Chemistry A*, 2022, 10 (43), 23037-23046?” A rough calculation shows that in a binary alloy acceptor system when the ratio of L8-BO: T9SBO-F is 1:0.2 will give a V_{oc} value of 0.876 V that is close to the measured V_{oc} value of 0.881V for the PM6:L8-BO: T9SBO-F ternary blend device. The 1-D WAXS curves also show similar pack features in the PM6:L8-BO and PM6:T9SBO-F systems in figure 4 g and i.

Response: Thank you for the suggestions. We have further fabricated devices with different T9SBO-F weight ratios and calculated the V_{oc} dependence on the ratio following the suggested work.^[1] The detailed device performance is provided in Fig. 1e and Table 1 in the supplementary information. As shown in Figure R10, the experimental results match well with the calculated V_{oc} at different ratios with mismatch less than 0.15%. Moreover, the ternary blend with enlarged domain size compared to binary films as shown in the GISAXS results (Figure R9) also supports possible alloy formation between L8-BO and T9SBO-F when blending them together. Corresponding revision has been made in the manuscript.

Figure R10. Voc dependency on the weight ratio of T9SBO-F in the acceptor. The value is obtained from at least 10 individual devices.

3. Third, from supplementary information, Figure 3b, the UV-Vis spectra show extended light absorption for the case of T9SBO-F film, is this also the reason that the EQE value is slightly larger in Figure 1?

Response: Thank you for the good comment. Indeed, T9SBO-F shows red-shifted UV-Vis spectrum. This is one of the reasons that the EQE value in 800-1000 nm enlarges when T9SBO-F content increases in Figure 1. Except for that, EQE is the ratio of the extracted free electron to the injected photons. On one hand, it is favorable to extend the absorption spectra of NFA to near infrared region. On the other hand, it is necessary to efficiently convert the absorbed photons into carriers, and extract the carriers from the device. Therefore, the extended absorption of T9SBO-F is a necessary but not sufficient condition for the improved integrated J_{sc} .^[2] For example, in Figure 1f, PM6:T9SBO-F shows lower EQE value in 650-800 nm, but higher EQE value in 800-100 nm. Combining EQE and absorption spectra, we are able to conclude that adding T9SBO-F into PM6:L8-BO shows not only extended absorption but also improved charge extraction efficiency. Corresponding revision has been made in the manuscript to acknowledge the contribution of extended absorption from T9SBO-F.

4. Fourth, Figure 1 should be revised to clearly demonstrate L8-BO and T9SBO-F molecular structures instead of just the core structures.

Response: Thank you for the advice, we have calculated the ESP of L8-BO and T9SBO-F molecules by Gaussian at B3LYP/6-31G (d, p) level (Figure R11). Both L8-BO and T9SBO-F almost show positive ESP distribution on the iso-surface around the backbone. L8-BO molecule presents a dipole moment of 0.57 D, whereas it is significantly increased to 3.26 D in T9SBO-F, which enables a stronger intra-molecular interactions.

Figure R11. The surficial electrostatic potential of the NFA molecules. (a) L8-BO and (b) T9SBO-F. The molecular geometries are optimized and ESP are calculated by Gaussian at B3LYP/6-31G (d, p) level.

References

1. Lin, Y.-C.; Chen, C.-H.; Lin, H.; Li, M.-H.; Chang, B.; Hsueh, T.-F.; Tsai, B. S.; Yang, Y.; Wei, K.-H., Binary alloy of functionalized small-molecule acceptors with the A–DA'D–A structure for ternary-blend photovoltaics displaying high open-circuit voltages and efficiencies. *Journal of Materials Chemistry A* **2022**, *10* (43), 23037-23046.
2. Cheng, H.-W.; Mohapatra, A.; Chang, Y.-M.; Liao, C.-Y.; Hsiao, Y.-T.; Chen, C.-H.; Lin, Y.-C.; Huang, S.-Y.; Chang, B.; Yang, Y.; Chu, C.-W.; Wei, K.-H., High-Performance Organic Solar Cells Featuring Double Bulk Heterojunction Structures with Vertical-Gradient Selenium Heterocyclic Nonfullerene Acceptor Concentrations. *ACS Applied Materials & Interfaces* **2021**, *13* (23), 27227-27236.

Reviewers' Comments:

Reviewer #1:

Remarks to the Author:

The authors fully answered for my comments. Therefore, I would like to recommend that this manuscript at the current stage can be published in Nature Communications.

Reviewer #2:

Remarks to the Author:

In this revised manuscript, the author made reasonable explanations to the questions raised. They systematically analysed the charge separation and recombination dynamics, quantified the related energy loss, and revealed their relationship with the molecular stacking and film morphology. The improved experiments and analysis can support the conclusion that the Se-substitution can improve ϵ_r of NFAs and further lead to an enhanced device performance. Hence, I am now thinking that this revised version can be published in Nature Communications without other revision.

Reviewer #3:

Remarks to the Author:

This reviewer is satisfied mostly with the revisions made by the authors except two minor points: there is a grammar error in Line 58 "which contributing to higher kinetic energy of the exciton/carrier, and thus accelerate exciton dissociation and charge." and there is a numerical error in Line 102 "The scale bar is from 1.28 eV (red) to 1.28 eV (blue)."

Reviewer #3:

This reviewer is satisfied mostly with the revisions made by the authors except two minor points: there is a grammar error in Line 58 “which contributing to higher kinetic energy of the exciton/carrier, and thus accelerate exciton dissociation and charge.” and there is a numerical error in Line 102 “The scale bar is from 1.28 eV (red) to 1.28 eV (blue).”

Response: Thank you for the careful and responsible reviewing. We have made revisions based on the reviewer’s suggestions. The sentences have been revised as follows:

1. “which contributing to higher kinetic energy of the exciton/carrier, and thus accelerate exciton dissociation and charge.” has been revised to “which **contributes** to higher kinetic energy of the exciton/carrier, and thus **accelerates** exciton dissociation and charge.”
2. “The scale bar is from 1.28 eV (red) to 1.28 eV (blue).” has been revised to “The scale bar is from **-1.28 eV (red)** to 1.28 eV (blue).”

Corresponding revisions have been included and highlighted in the manuscript.